# Advanced Multiplex Loop Mediated Isothermal Amplification (mLAMP) Combined with Lateral Flow Detection (LFD) for Rapid Detection of Two Prevalent Malaria Species in India and Melting Curve Analysis

**DOI:** 10.3390/diagnostics12010032

**Published:** 2021-12-24

**Authors:** Supriya Sharma, Sandeep Kumar, Md Zohaib Ahmed, Nitin Bhardwaj, Jaskirat Singh, Sarita Kumari, Deepali Savargaonkar, Anupkumar R. Anvikar, Jyoti Das

**Affiliations:** Parasite Host Biology, ICMR-National Institute of Malaria Research, New Delhi 110077, India; supsmicro@gmail.com (S.S.); sandeepbt81@yahoo.com (S.K.); zohaiba01@gmail.com (M.Z.A.); nitinbh529@gmail.com (N.B.); jaskiratsinghrishi@gmail.com (J.S.); ssarita.pat@gmail.com (S.K.); dr.deepali27@gmail.com (D.S.)

**Keywords:** malaria, diagnosis, multiplex loop-mediated isothermal amplification (mLAMP), lateral flow detection (LFD), point of care (POC)

## Abstract

Isothermal techniques with lateral flow detection have emerged as a point of care (POC) technique for malaria, a major parasitic disease in tropical countries such as India. *Plasmodium falciparum* and *Plasmodium vivax* are the two most prevalent malaria species found in the country. An advanced multiplex loop-mediated isothermal amplification (mLAMP) combined with a lateral flow dipstick (LFD) technique was developed for the swift and accurate detection of *P. falciparum* and *P. vivax*, overcoming the challenges of the existing RDTs (rapid diagnostic tests). A single set of LAMP primers with a biotinylated backward inner primer (BIP primer) was used for DNA amplification of both malaria species in a single tube. The amplified DNA was hybridized with fluorescein isothiocyanate (FITC) and digoxigenin-labelled DNA probes, having a complemented sequence for the *P. falciparum* and *P. vivax* genomes, respectively. A colour band appeared on two separate LFDs for *P. falciparum* and *P. vivax* upon running the hybridized solution over them. In total, 39 clinical samples were collected from ICMR-NIMR, New Delhi. Melting curve analysis, with cross primers for both species, was used to ascertain specificity, and the sensitivity was equated with a polymerase chain reaction (PCR). The results were visualized on the LFD for both species within 60 min. We found 100% sensitivity and specificity, when compared with a traditional PCR. Melting curve analysis of mLAMP revealed the lowest detection limit of 0.15 pg/μL from sample genomic DNA. The mLAMP-LFD assays could be a potential point of care (POC) tool for early diagnosis in non-laboratory conditions, with the convenience of a reduced assay time and the simple interpretation of results.

## 1. Introduction

The World Health Organization (WHO) calculated that 229 million cases of malaria occurred globally in the year 2019. In the South East Asia (SEA) region, the occurrence of *P. falciparum* (*Pf*) and *P. vivax (Pv)* was 50% and 53%, respectively, while *Pv* incidence in India was 47% [1]. Accordingly, *Pf* and *Pv* were roughly the most prevalent *Plasmodium* parasite species causing malaria in India. Prompt and accurate diagnosis of the disease is important as this determines the treatment guidelines; prompt and accurate diagnosis is the backbone of disease remedy. Malaria is diagnosed classically by microscopy, which is considered the gold standard [2], and the other leading method used is rapid diagnostic tests (RDTs). Molecular diagnosis of malaria has many paradigms, including polymerase chain reaction (PCR), quantitative PCR (qPCR) or reverse transcriptase PCR (RT-PCR), semi-nested PCR (n-PCR), multiplex PCR (m-PCR), and isothermal methods, namely, loop-mediated isothermal amplification (LAMP), recombinase polymerase amplification (RPA), etc. Microscopy requires trained workers, is tedious, and is often found to be less sensitive, when compared to other high-end methods [3]. RDTs work effectively as they are quick and give reliable results, but the *Pfhrp*2 gene deletion problem has raised a query on its use. The debate continues over the judicious use of RDTs based on the HRPII antigen [4].

Nested PCR remains the highly preferred tool, when compared with RDTs or microscopy, but it is time-consuming, as it requires almost a day or more to identify a sample for all five human malaria-causing *Plasmodium* species, which is tedious and increases the total cost of diagnosis [5]. To overcome this issue, multiplex PCR is used, which not only amplifies more than one target at a time but also uses fewer chemicals and consumables, even though it requires target-specific primers and defined protocols [6]. qPCR requires a sophisticated machine and a well-trained individual to operate and interpret the data, thus making it unsuitable for the routine diagnosis of malaria [7]. In addition, there are other molecular techniques for malaria parasite detection such as magnetic resonance, as well as detection strategies with instruments, such as the relaxometer [8,9,10,11,12,13], the blue ray optical device [14], rotating-crystal magneto-optical detection (RMOD) [15], and the photoacoustics (PA) excited surface acoustic wave (SAW) sensor [16]. These techniques were demonstrated at various levels of proof of concept (e.g., animal models), but routine human diagnosis reports have yet to be seen. LAMP is an isothermal process that has added a new avenue of diagnosis for malaria. Amplification of genomic DNA/RNA under a constant temperature is facilitated by loop primers, which are designed with the utmost care [17]. However, there are many reports, which indicate specificity issues in the reaction. A false-positive result is common when there is contamination in the laboratory or of the chemicals. The challenge of designing a multiplex primer for LAMP is increased due to the fact that one set of LAMP primer contains six primers, meaning a multiplex primer needs more sets. A multiplex primer could also increase the overall amount of primer in the total reaction volume, which may cause misdiagnosis. The design in this study focuses on a single set of LAMP primers containing four primers and two species-specific probes labelled with fluorescent dyes to counteract the overcrowding of primers in the advanced multiplex-LAMP (mLAMP) reaction. This LAMP primer has a biotinylated BIP primer for the DNA amplification of both malaria species. Subsequently, the amplified DNA was hybridized with fluorescein isothiocyanate (FITC) and digoxigenin-labelled DNA probes, which have complement sequences for the *Pf* and *Pv* genomes, respectively. Later, the hybridized solution was run on two separate lateral flow dipsticks, which show a colour band for *Pf* and *Pv.* This approach not only enhanced the specificity of the method but also allowed naked eye visualization. Although LAMP reactions are visualized by the naked eye using various dyes, such as SYBR green, calcein, HNBs, etc., along with agarose gel electrophoresis, multiplex LAMP reactions have some pros and cons [6]. In a multiplex LAMP reaction, it is not easy to distinguish the ladder pattern of two different species.

Isothermal methods have emerged as a naive tool for malaria diagnosis. Combining isothermal methods with a lateral flow device has introduced a new era of point of care (POC) diagnosis [18]. The aforementioned combination was utilized to design mLAMP with a lateral flow dipstick (mLAMP-LFD). The multiplex primers were designed using the18S SSU rRNA gene target region of *Plasmodium*, and the visualization was accomplished using the biotin-FITC and biotin-digoxigenin complex on the lateral chromatographic strips. The specificity of this new method was assessed by melting curve analysis with cross primers for both species. The different concentrations of primers, different concentrations of genomic DNA, and mixing of species primers were used for determining specificity. The sensitivity of these data was calculated using 39 clinical samples from the malaria clinic of ICMR-NIMR, New Delhi. There are already a few commercial LAMP products available on the market for malaria diagnosis, such as Illumigene and Loopamp. There were eight studies conducted in sub-Saharan Africa, which applied commercial LAMP kits as a screening tool in malaria prevalence surveys [19] These commercial kits are not readily available in India and are of high price. Illumigene^®^-Pan-Plasmodium is priced at EUR 28, which is approximately INR 2403.38, and LoopampTM-Pan-Plasmodium for detection of *Pf and Pv*, is priced at EUR 5.2, which is approximately INR 429.10 [20]. In India, no such study has been conducted. This work will lead to a commercial kit with an indigenous primer, which will be comparatively lower in price. Both these brands target the mitochondrial DNA of *Plasmodium* species detection; however, for genus-level identification 18s ribosomal RNA is considered better as it has a conserved region. Mitochondrial target is used for pan detection as it is conserved for the *Plasmodium* genus, but 18sRNA can easily differentiate between species.

The new POC mLAMP-LFD of the present study may assist in meeting the new diagnostic need of malaria diagnosis and treatment strategies. It may also help in identifying unnoticed mixed infections and the exact assessment of species-specific malaria prevalence. It could be an aggressive tool for reaching the malaria elimination goal in the near future.

## 2. Materials and Methods

### 2.1. Study Samples

The study was carried out at the ICMR-National Institute of Malaria Research, New Delhi. The blood samples of fever patients suspected of malaria were collected from the Malaria Clinic of ICMR-NIMR, after obtaining consent from them, by a single finger prick on filter paper (Whatman 3 × 3 mm). The blood was used for microscopy (8 μL), RDT (5 μL), PCR (50 μL), and LAMP (50 μL)-based experiments. A total of 39 clinical samples were included in the study for assessing the sensitivity of the method. Control samples with known parasitaemia were collected from the WHO Malaria NAAT EQA scheme, which includes different parasite density dried blood spot samples from all human species of malaria, with positive and negative samples.

### 2.2. Microscopy and Rapid Diagnostic Test (RDT)

The thick and thin smear was prepared and stained with JSBI and JSBII stains and observed under an oil immersion microscope. SD BIOLINE Malaria Ag *P.f*/*P.v* test (Standard Diagnostics Ref 05FK80) was used for screening all samples in accordance with the manufacturer’s instructions. The patients who tested positive for malaria were given treatment as per the national treatment guidelines. The results of the microscopy and RDTs were compared with the mLAMP result.

### 2.3. DNA Isolation for Polymerase Chain Reaction (PCR) and Loop-Mediated Isothermal Amplification (LAMP)

A commercial blood DNA extraction kit (QIAamp DNA Mini Kit, QIAgen, Catalog #51304) was used for parasite DNA extraction from the dried blood spot using the manufacturer’s instructions. In short, 180 μL of lysis buffer was added to a microcentrifuge tube containing three punches of the dried blood spot, and it was incubated at 85 °C with shaking for 60 min. Proteinase K (20 μL) was added to the lysate and was centrifuged at 14,000× *g* then incubated at 56 °C for 60 min. In elution solution, the DNA was eluted (150 μL) and quantified spectrophotometrically (NanoDrop 2000 Spectrophotometer-Thermo Scientific, Waltham, MA, USA).

### 2.4. PCR

The isolated DNA was amplified using the mitochondrial (mt) target to amplify the *Plasmodium* genus using published primers [21] to verify the presence or absence of malaria infection based on Plasmodium DNA. The result of the mt PCR was compared with the result of the LAMP reaction for the genus. PCR was performed on all 39 samples with 2 μL of DNA template extracted from the blood spot. The cycle condition was started with an initial denaturation at 94 °C for 5 min, followed by 50 cycles at 94 °C for 30 s, 55 °C for 30 s, 72 °C for 1 min, and a final hold at 4 °C. The PCR products were run on a 1.5% agarose gel at 100 volts for 40 min. The results were visualized under the gel documentation system in UV light (Alpha Innotech, Merced St. San Leandro, CA, USA).

### 2.5. LAMP Primers

The18S SSU rRNA gene target region of Plasmodium was selected as a target for this method. A set of six primers were designed using Primer Explorer version 5 to target the species-specific region of *Pf* (DNA Data Bank of Japan Accession No. LC483576) and *Pv* (DNA Data Bank of Japan Accession No. LC483577), as shown in Table 1. The backward inner primer of the genus primer was labelled with biotin. Probes for *Pf* and *Pv* were designed and labelled with digoxigenin and FITC. The oligonucleotides were commercially synthesized at Eurofins Genomics India Pvt Ltd. (Bangalore, India). The LAMP products were visualized by performing agarose gel electrophoresis.

### 2.6. Reaction Mixture for LAMP

mLAMP was carried out in a total of 25 μL of a reaction mixture containing 8 U of Bst DNA polymerase, 2.5 mL 10 X Thermo Pol Buffer, 4 mM MgCl_2_, 1 mM dNTPs, 1.6 mM each of FIP and BIP, 0.2 mM each of F3 and B3, 0.94 M betaine, 150 mM SYBR green, and 2 μL of target DNA (5 ng). The reactions were performed in 0.2 mL centrifuge tubes in a thermal cycler machine for temperature control. LAMP reactions were performed under the isothermal condition of 63 °C for 25 min and 85 °C for 5 min to stop the reaction. A typical ladder-like pattern on 2% gel electrophoresis indicated a positive LAMP reaction. The reactions were run in duplicate to ascertain the accuracy of the results.

### 2.7. Melting Curve Analysis and Specificity Assessment

Melting curve analysis of 39 clinical samples was performed on a qPCR machine (Light Cycler 480, Roche Rotkreuz, Switzerland). qPCR was performed in a 96-well plate (Bio-rad, Hercules, CA, USA). The reaction volume was 20 μL. It contained 2X SYBR green PCR master mix (KAPA Biosystems, Wilmington, MA USA), 0.25 μM of each primer, cDNA, and water to adjust the volume. The experiments were completed by repeating twice. As an endogenous control, the Human Beta-actin gene was used with primers and cycling conditions, as described in [22]. Healthy control samples were used as calibrators, while a 3D7 strain of *Pf* and a confirmed field sample of *Pv* with genomic DNA concentration 5 ng/μL were used for standardization purposes. The relative fold changes were calculated using the 2^-∆∆CT^ method [23]. The specificity of the *Pf* and *Pv* primers was ascertained by using the same set of serially diluted templates in cross-reaction and also by involving the non-target templates of other *Plasmodium* species.

### 2.8. Construction of Lateral Flow Strip

The lateral flow strip was prepared using an Easy pack Membrane Kit (MDI membrane technology, Menesar, Gurugram, India). This kit consists of various laminates of different pore size membranes, absorbent pads, sample pads, and release matrices whose optimal combination for the performance of the test was achieved after many trials. The best combination we found was with the use of an absorbent pad of size 22 × 260 mm (Type: AP 110), nitrocellulose membrane of high protein binding capacity size 60 × 260 mm (Type: CNPC-SS12-L2-P25), release matrix of size 70 × 260 mm (Type: PT-R7), and sample pad of size 14 × 260 mm (Type: GFB-R4 (0.35). The assembled components were cut into thin strips of size 0.3–0.5 cm in width and 6 cm in length for lateral flow as shown in Figure 1. The final strip was coated with 1 μL of 2 mg/mL stock solution goat anti-mouse Ab (Cat No-AP124, Merck, Darmstadt, Germany) for the control line and with 1 μL of 1 mg/mL stock solution Streptavidin (Cat no-85878, Sigma Aldrich, Darmstadt, Germany) for the test line. This, together with a commercial lateral flow strip (PCRbeam™ Fast PCR Detection Kit, highQu GmbH, Kraichtal, Germany), was used as control strips to check the accuracy of both the hybridized products and their running solutions.

#### 2.8.1. Functionalization of Gold Nanoparticles

The gold nanoparticles of 20 nm diameter (Cat No- 741965, Merck, Darmstadt, Germany) were used as signal generators, which were conjugated with mouse anti-FITC antibody (Cat No- SAB4200738, Roche, Rotkreuz, Switzerland) for *Pv* and anti-digoxigenin (Cat No- 11333062910, Roche, Rotkreuz, Switzerland) antibody for *Pf* (Roche, Rotkreuz, Switzerland) detection. The protocol described previously was used [24]. Briefly, the pH of 10 mL colloidal gold nanoparticles was adjusted to 8, and 22 μg of anti-FITC Ab was added gently at room temperature for 1 h. BSA 1% (*w*/*v*) was added to this solution and kept for another 1 h. Finally, the mixture was centrifuged at 10,000× *g* for 30 min at 4 °C. The pellet was suspended in 10 mM phosphate buffer, which contained 0.05% (*w*/*v*) NaN_3_ and 1% (*w*/*v*) BSA.

#### 2.8.2. Hybridization Buffer Composition and Hybridization Protocol

To the PCR product, an equal volume of 10% (*w*/*v*) polyethylene glycol 3350 (PEG 3350) in 1.5 M MgCl_2_ was mixed by simple pipetting. The hybridization of the LAMP amplified product and the labelled probe was carried out in a 0.2 mL tube. In this, 2 μL of the labelled probe, hybridization buffer (20 μL), and 5 μL of the amplified LAMP product with biotin tagged in the BIP primer were mixed and preheated at 95 °C for 10 min in a thermal cycler (ABS Veriti, Thermo Fisher Scientific, Waltham, MA, USA) and hybridized at 54 °C for 10 min hold at infinity at 4 °C.

#### 2.8.3. Visualization on a Lateral Strip

The hybridized LAMP product (50 μL), functionalized gold nanoparticles (10 μL), and 50 μL of running buffer (0.01% Triton X-100 (*v*/*v*) in 2 mM of phosphate buffer) were mixed in a 0.2 mL tube, and the lateral flow strip was dipped in it from the sample pad side and kept for 15 min for visualization, Figure 2.

### 2.9. Statistical Analyses

The statistical software Statistical Package for Social Science (SPSS) (IBM SPSS Statistics for Windows, Version 17.0. Armonk, NY, USA: IBM Corp) was used for statistical analyses. The data were presented as no. (%) or mean ± SD/median (Min, Max). The baseline characteristics were compared between the groups using the chi-square test for categorical variables and the Mann–Whitney U test for continuous variables. The results were reported as a difference in proportion (95% CI). *p*-value < 0.05 was considered as statistically significant.

### 2.10. Ethics

This study is part of a project entitled “Development of simple and accurate point of care malaria diagnostic method for rapid detection and identification of five human *Plasmodium* spp. using mloop-mediated isothermal amplification (LAMP)”, which has been approved by the Institutional Ethics Committee of ICMR-NIMR, New Delhi (letter number-ECR/NIMR/EC/2018/39, dated 26 February 2018).

## 3. Results

### 3.1. Microscopy, RDT, Mitochondrial PCR (Mt PCR) and LAMP Result

The 39 clinical samples were subjected to malaria testing by a gold standard microscopy test in which 20 and 19 samples were found to be positive and negative, respectively, for malaria, followed by RDT, which gave 21 positives and 18 negatives. The genomic DNA was isolated and two molecular tests were performed to quantify the number of positives and negatives by the standard PCR method and by newly developed LAMP primer having BIP-biotin primer. Both tests amplify the genus *Plasmodium*. Mutually the tests gave similar results with 26 positive and 13 negative malaria samples (Table 2).

### 3.2. Optimizing the LAMP Assay Conditions and Its Sensitivity

The LAMP primers were standardized for the time of amplification, and the minimum time for amplification found was 15 min, as one can observe in Figure 3a; however, we propose that, to confirm a true positive amplification, the optimal time should be 20–25 min. A 45-min LAMP reaction at 59–69 °C generated many ladder-like pattern bands of various brightness values when visualized by 1.5% agarose gel electrophoresis (Figure 3b) Since LAMP generated more distinct and brighter bands at 63 °C, we would advocate that 63 °C is the optimal temperature for amplification. The distinguishing band after amplification was achieved at 0.15 pg/μL in a reaction run for 45 minutes at 63 °C with different concentrations of DNA ranging from 5 nanograms (ng) to 5 femtograms (fg) (Figure 3c), suggesting these primers can be used for detecting the infection of low parasite density.

### 3.3. Sensitivity with Other Standard Methods

The study data were analysed in two ways to attain the sensitivity of the method statistically. The classical diagnostic method of microscopy was first taken as the gold standard, and the data were compared with RDT, PCR, and LAMP. The comparison yielded 95%, 100%, and 100% sensitivity as well as 89.47%, 68.42%, and 68.42% specificity, respectively (Table 3A). However, when the molecular method using PCR was taken as the gold standard and compared with the RDT, LAMP, and microscopy, the result was 80.77%, 100% and 76.92% sensitivity and 100%, 100% and 100% specificity, respectively (Table 3B). Moreover, diagnostic methods’ variability was statistically checked among the methods keeping microscopy as the gold standard to the RDT, mt PCR, and LAMP methods. The LAMP method showed 20 malaria positives, when compared with microscopy. To predict the cases with the existing gold standard marker microscopy and PCR, the receiver operating characteristic curve (ROC) analysis was performed. In this study, it was found that all markers had a good score for prediction; RDT had 92.2% area under the curve (AUC), PCR had 84.2% AUC, while LAMP had 84.2% AUC (Figure 4A). Interestingly, with the existing gold standard marker-PCR, the study revealed that RDT had 90.4% area under the curve (AUC), LAMP had 100% AUC, and microscopy had 88.5% AUC (Figure 4B).

### 3.4. Melting Curve Analysis for Specificity Determination

The standard DNA of the 3D7 strain of *Pf* with a calculated DNA concentration of 5 ng was diluted in molecular grade water until 5 fg concentrations, and similarly, a clinical sample of *Pv* with a DNA concentration of 5 ng was diluted up to 5 fg. Both these serial dilutions were subjected to an ideal condition of primer concentration of 250 ng/reaction. The malaria LAMP system was able to detect *Pf* at a concentration up to 5 fg with the given 250 ng concentration of primers within 90 min with standard amplification cycles and SYBR green dye. However, with the same concentration of 250 ng of primers it was able to detect up to 10 fg concentration of *Pv* DNA with the standard cyclic conditions (Figure 5a,b). The sensitivity of these LAMP primers is seen as the shape peak obtained at different concentrations of the diluted genomic DNA of the malaria parasite.

The *Pf* and *Pv* LAMP primers were observed specifically for their respective genomic DNA as in the cross-reactive reactions, and no amplification and no melting curve were found for cross-species as well as for other human pathogenic malaria species.

### 3.5. Principle and Result of mLAMP-LFD (Lateral Flow Detection)

The mLAMP-LFD strip consisted of a chromatography control line (CL) and a test line (TL). The CL has a goat anti-mouse antibody that captures the gold nanoparticle conjugated with the anti-FITC and anti-digoxigenin antibody, whereas the test line has streptavidin as its base which captures the biotin-labelled LAMP amplified products with biotin labelled pan primer. The test was considered valid when the CL line showed the signal; however, when there was no signal on the test line, it implied an invalid test. Similarly, in the PCRbeam™ Fast PCR Detection Kit, the test line and control line were interpreted. The multiplexing of *P. falciparum* and *P. vivax* was achieved successfully. Both of the species were easily identified on the lateral flow strip (PCRbeam™ Fast PCR Detection Kit) as well as on the assembled lateral flow strip after amplification using the target primers and probes (Figure 6a,b).

## 4. Discussion

India has a target for elimination of malaria by the year 2030. It is evident that new diagnostics may accelerate its fight against malaria, as the reported cases of HRP2 deletions are also rising slightly [4]. Moreover, the asymptomatic cases and mixed infections add to the challenges of elimination. There is a report that shows the importance of identifying *Pf* and *Pv* and their mixed infection in India [25]. LAMP delivers an opportunity for enhanced field-friendly diagnosis of malaria infections in endemic areas [17]. The clinical sensitivity and specificity of LAMP were evaluated by comparing it with the conventional methods of microscopy, RDTs, and PCR, and for the detection of two *Plasmodium* species in human samples. This was a laboratory-scale pilot testing of the method. The statistically significant sample size for biological samples with diagnosis report pilot testing is 30, and we conducted a study on 39 samples as per the availability of samples. The optimal time was selected as 20–25 min, as it could allow sufficient assessment of the samples and showed amplification within a short time. The optimal temperature selected was 63 °C as we were able to obtain a higher quantity of DNA. At 67 °C, it showed a more distinct and ladder-like pattern, as the amplification was less meaning a low quantity of amplified DNA was present and, hence, no smearing. Moreover, it is easier to maintain a 5 °C lower temperature of 63 °C than 67 °C and it is reached in less time. The sensitivity and specificity of the method were found to be 100%, which is in accordance with the previously published data [26]. It is a known fact that there are high chances for erroneous results when the LAMP products are exposed to gel electrophoresis due to rigorous contamination by dispersal of LAMP products. It is difficult in mLAMP products to evaluate two or more different species in multiple bands. Therefore, a few authors have suggested that melting curve analysis should be there to examine the non-specific amplification [27]. The melting curve analysis has been used for the identification of malaria species [28] and by the LAMP method in other species [29]. However, here we not only used the melting curve for diagnosis but also to examine the specificity of the method. Nevertheless, LAMP techniques have disadvantages due to their high amplification sensitivity and the fact that cross-contamination is unavoidable. These shortcomings can be rectified. Good laboratory practices, including the sterilization of the bench, pipettes, and other in-use items before preparing the LAMP reaction, can satisfactorily minimize contamination.

A dipstick is a handy, low-cost, easy to use point of care device. The basic principle of lateral immunoassay with antibody–antigen interactions is used with gold nanoparticles for visualization by the naked eye. This certainly abolishes the need for machinery and skilled individuals for analysis of the result [30]. The results are generated within an hour. There is no health hazard associated with these devices, as no use of carcinogenic EtBr and UV rays are involved, and less waste is generated [24].

There are reports in which lateral flow devices were combined with LAMP reactions to detect malaria species [19,31,32,33]. These reports have shown the different target genes and reactions conditions for detection of malaria. This study advances from these previous studies, as the number of primers used in the study to identify *Pf* and *Pv* was reduced, which leads to more targeted amplification and lessens the chances of false negative results. The high sensitivity of 0.15 pg/μL was an add-on for POC diagnosis.

mLAMP-LFD could be a consistent, convenient, and possible substitute for molecular diagnosis of *Plasmodium* infection, particularly in high malaria transmission regions. It is time-saving and of low cost, and the easy interpretation of results makes it a potentially useful tool for POC diagnosis. However, this LAMP assay needs a hybridization step for species identification.

## 5. Conclusions

The advanced mLAMP-LFDassay could rapidly amplify DNA targets of *Pf* and *Pv* simultaneously under isothermal temperature with visualization by the naked eye. The high sensitivity and specificity boost confidence in this assay as a POC method, which may aid in efforts to reach the malaria elimination goal.

## Figures and Tables

**Figure 1 diagnostics-12-00032-f001:**
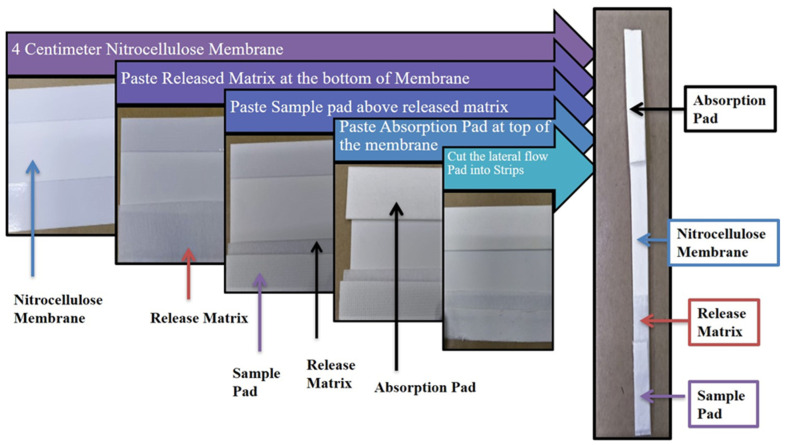
Assembly of membranes of Easy pack Membrane Kit components (MDI membrane technology, Gurugram, India) into lateral flow strip.

**Figure 2 diagnostics-12-00032-f002:**
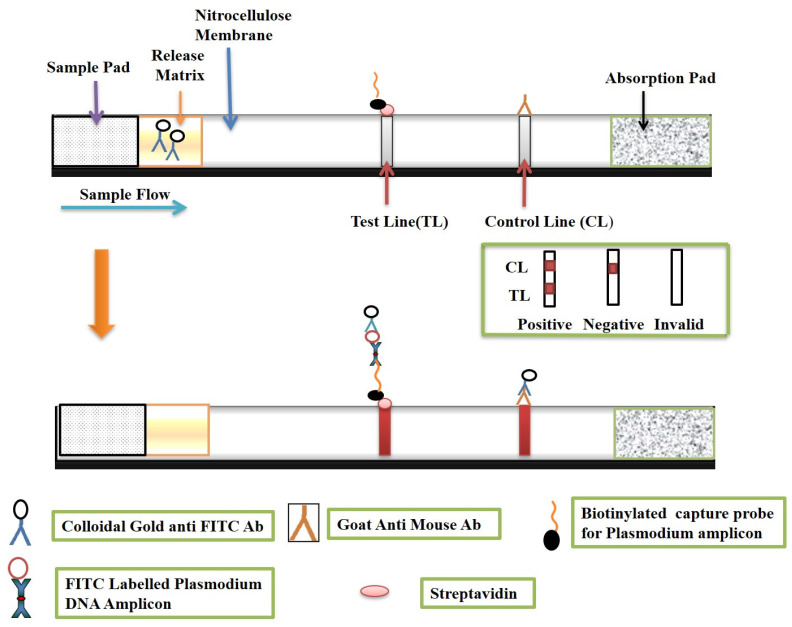
Schematic representation of lateral flow strip principle and its visualization.

**Figure 3 diagnostics-12-00032-f003:**
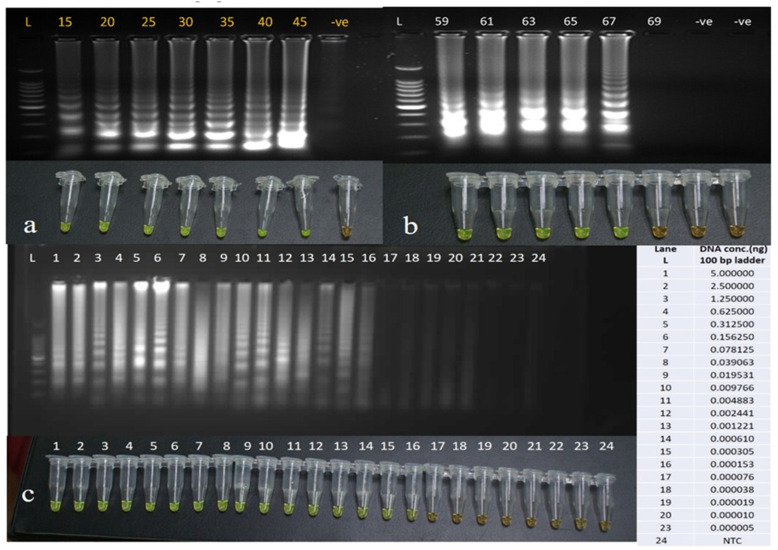
(**a**) Gel image and tubes of LAMP reaction showing naked eye visualization with SYBR green for standardization of time for amplification. (**b**) Gel image and tubes of LAMP reaction showing naked eye visualization with SYBR green for the standardization with different temperatures ranging from 59 to 69 °C. (**c**) Gel image and tubes of LAMP reaction showing visualization with the naked eye with SYBR green for standardization with different concentration of genomic DNA ranging from 5 ng to 0.000005 ng. L- 100 base pair ladder/marker, -ve–Negative control.

**Figure 4 diagnostics-12-00032-f004:**
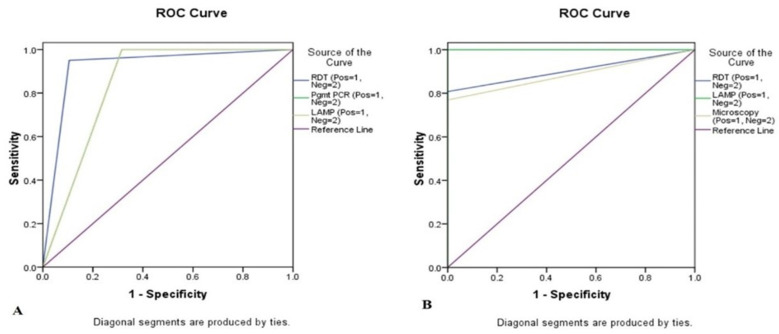
Receiver operating characteristic (ROC) curve analysis data showing prediction of case by the methods (**A**) when microscopy is the gold standard and (**B**) when PCR is the gold standard.

**Figure 5 diagnostics-12-00032-f005:**
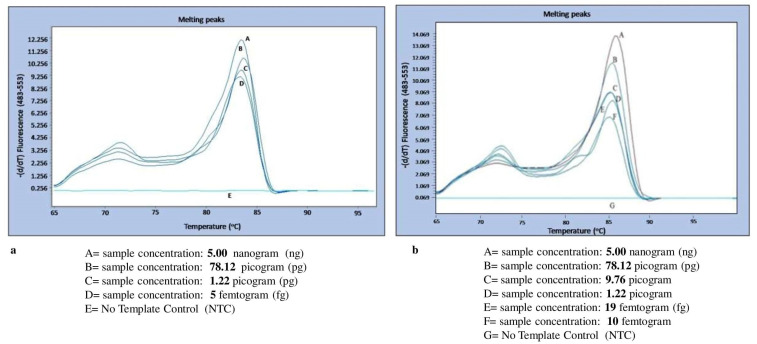
(**a**,**b**): Melting curve showing the detection limit for *P. falciparum* and *P. vivax* with respect to constant primer concentration (250 ng) and different sample concentrations. The graph was generated by using Light Cycler 480 software (version 4., Light Cycler 480, Roche Rotkreuz, Switzerland).

**Figure 6 diagnostics-12-00032-f006:**
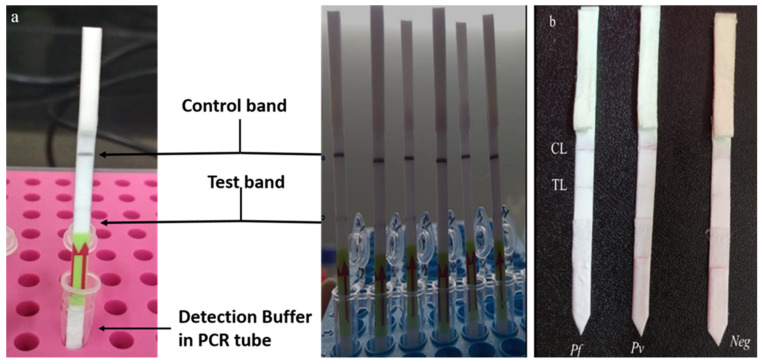
(**a**,**b**)**:** PCRbeam™ fast polymerase chain reaction (PCR) detection dipstick in PCR tube with detection buffer showing control band and test band. 2. *P. vivax* (*n* = 2), control sample (*n* = 2), and no template (*n* = 2) on 6 strips from left to right. When there was no PCR product in the reaction, then only the control band was visible. When the target specific product was present, the test band was visible. Similarly, we found the results for *P. falciparum* and *P. vivax* on the assembled lateral flow strips as shown in (**b**). CL— Control line and TL—Test line.

**Table 1 diagnostics-12-00032-t001:** The primer sequence for amplification of the *Plasmodium* genus and species using the loop-mediated isothermal amplification (LAMP) reaction is as shown below.

Primer	Sequence
F3	TCCATTAATCAAGAACGAAAGT
B3	CCCAGAACCCAAAGACTT
FIP	TCCAAAGCCTAGTCGGCATAGAGGGAGTGAAGACGATCAGATACCGTCGTAAT
BIP-biotin	GAGTTTTTCTTTTCTCTCCGGAGA-TTCTCATAAGGCACTGAAGG
*Pf* Digoxigenin probe	AAGTCATCTTTCTAGGTGACT
*Pv* FITC probe	GAATTTTCTCTTCGGAGTTTATT

**Table 2 diagnostics-12-00032-t002:** Summary of performance of microscopy, rapid diagnostic test (RDT), mitochondrial polymerase chain reaction (PCR), and LAMP methods in clinical samples.

Method (*n* = 39)	RDT	Microscopy	Mt PCR	LAMP
**Positive**	21	20	26	26
**Negative**	18	19	13	13

**Table 3 diagnostics-12-00032-t003:** Sensitivity and specificity of LAMP compared with microscopy, RDT, and mt PCR. (**A**) Representing data when microscopy is treated as the gold standard, and (**B**) when the molecular method PCR is treated as the gold standard.

A	Microscopy Positive (*n* = 20)	Microscopy Negative (*n* = 19)	Sensitivity% (95% C.I)	Specificity% (95% C.I)	PPV% (95% C.I)	NPV% (95% C.I)
RDT Positive	19 (95.0%)	2 (10.5%)	95 (75.13–99.87)	89.47 (66.86–98.70)	90.48 (71.84–97.25)	94.44 (71.44–99.14)
RDT Negative	1 (5%)	17 (89.5%)				
Mt PCR Positive	20 (100%)	6 (31.6%)	100 (83.16–100.00)	68.42 (43.45–87.42)	76.92 (63.23–86.60)	100
Mt PCR Negative	0 (0%)	13(68.4%)				
LAMP Positive	20 (100%)	6 (31.6%)	100 (83.16–100.00)	68.42 (43.45–87.42)	76.92 (63.23–86.60)	100
LAMP Negative	0 (0%)	13(68.4%)				

**B**	**Mt PCR** **Positive** **(*n* = 26)**	**Mt PCR** **Negative** **(*n* = 13)**	**Sensitivity%** **(95% C.I**)	**Specificity%** **(95% C.I)**	**PPV%** **(95% C.I**)	**NPV%** **(95% C.I)**
RDT Positive	21 (80.8%)	0 (0%)	80.77 (60.65- 93.45)	100 (75.29–100.00)	100	72.22 (54.18–85.11)
RDT Negative	5 (19.2%)	13 (100%)				
LAMP Positive	26 (100%)	0 (0%)	100 (86.77–100.00)	100 (75.29–100.00)	100	100
LAMP Negative	0 (0%)	13 (100%)				
Microscopy Positive	20 (76.9)	0 (0%)	76.92 (56.35–91.03)	100 (75.29–100.00)	100	68.42 (51.78–81.38)
Microscopy Negative	6 (23.1%)	13 (100%)				

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
