# Peer review of "Advanced Multiplex Loop Mediated Isothermal Amplification (mLAMP) Combined with Lateral Flow Detection (LFD) for Rapid Detection of Two Prevalent Malaria Species in India and Melting Curve Analysis"

_diagnostics, 2021, doi:10.3390/diagnostics12010032_

Round 1
Reviewer 1 Report
Please see the attached file.

Reviewer 2 Report
In this study, authors standardized isothermal method (LAMP) as a malaria diagnostic kit. They combined isothermal method-LAMP with lateral flow device. Authors designed multiplex primers using the 18S SSU rRNA gene target region of Plasmodium, followed by visualization using Biotin-FITC and Biotin-Digoxigenin complex on the lateral chromatographic strip. Using the mLAMP-LFD method, authors studied 39 clinical samples and results were compared with other existing methods. This study is advance from the rest of the previous studies by mainly two ways- sensitivity and reduced primer number. Some of the minor comments are-
- Please make sure spacing between words are correct. One word spacing is missing in page 4 and line 146, e.g. wascarried.
- What is againg in page 10 and line 311. Is it against??
- Specify the DNA concentration used in material and methods 2.6.
Round 2
Reviewer 1 Report
Please see the attached file.
